# MULTI-CHANNEL GRAPH CONVOLUTIONS

## ABSTRACT

Defining the convolution on graphs has led to much progress in graph machine learning, particularly through approximations based on polynomials and, ultimately, message-passing neural networks (MPNNs). However, this convolution is defined for single-channel graph signals, i.e., a single feature is given at each node, and a single new feature is assigned to each node. As multiple initial node features are provided for many challenging tasks and convolutions are generally defined for these multi-channel signals, we introduce multi-channel graph convolutions (MCGCs) by obtaining their form using the graph Fourier transform. MCGCs highlight the critical importance of utilizing multiple edge relations to amplify different signals for each feature channel. We further introduce localized multi-channel MPNNs and the multi-channel graph isomorphism network (MC-GINs), with which we can provably obtain linear mappings that are injective on multisets. Our experiments confirm the greatly improved capabilities of MCGCs and MC-GINs.

## 1 INTRODUCTION

Many challenging tasks and applications are based on graph-structured data, e.g., property prediction for molecules (Hu et al., 2021), fraud detection in transactions (Weber et al., 2019), and recommendations (Monti et al., 2019). Developing expressive and well-performing methods to learn from such data is thus an important challenge. As one such method, neural networks for graph-structured data are based on defining a convolution on a graph. Spectral graph convolutions emerged by computing the convolution exactly in the graph Fourier domain based on the convolution theorem (Hammond et al., 2011; Bruna et al., 2014). Due to its high computational cost, various approximations based on polynomials of the graph Laplacian emerged (Levie et al., 2019; He et al., 2021; Koke & Cremers, 2024), e.g., using Chebyshev polynomials (Defferrard et al., 2016). This further led to the graph convolutional network (GCN) (Kipf & Welling, 2017) as a localized approximation. Most currently used message-passing neural networks (MPNNs) are derivations and improvements of this spectral graph convolution and the GCN. This includes applying different aggregation functions like the mean (Hamilton et al., 2017), the sum (Xu et al., 2019), utilizing attention coefficients (Velickovic et al., 2018; Brody et al., 2022) or negative edge weights (Yan et al., 2022). More complex methods like gating mechanisms (Li et al., 2016; Rusch et al., 2023), positional encodings (Kreuzer et al., 2021; Rampásek et al., 2022; Huang et al., 2024), and normalization layers (Zhao & Akoglu, 2020) are typically combined with these approximated graph convolutions.

However, this convolution is defined for single-channel signals, i.e., each node has a single feature and a single feature per node is obtained. In most applications, each node has multiple initial features assigned to it, e.g., a text embedding for documents and atom features for molecular data. Similarly, the goal of these models is often to find rich node embeddings capturing both structural properties and feature interactions, which are typically designed to have multiple feature channels. The currently used definition of graph convolutions and, consequently, most MPNNs are ill-defined for this task. As MPNNs amplify the same signal for each channel, issues like representational rank collapse (Roth & Liebig, 2023) and over-smoothing (Oono & Suzuki, 2020) emerged.

The convolution that is mapping multi-channel signals to multi-channel signals is generally defined differently. We introduce these multi-channel convolutions to graphs by obtaining their form based on the convolution theorem and the graph Fourier transform. This multi-channel graph convolution (MCGC) highlights the importance of utilizing multiple edge relations. Each edge relation corresponds to a different signal, and the feature transformation describes the amplification or damping

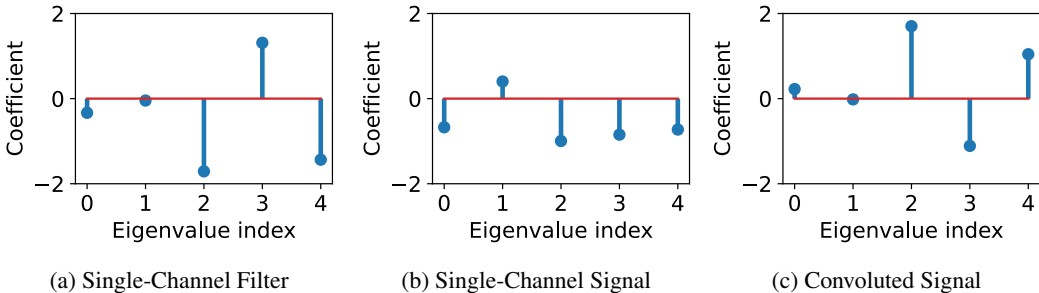

(a) Single-Channel Filter      (b) Single-Channel Signal      (c) Convoluted Signal

Figure 1: Single-Channel signal and filter in the Fourier domain. The eigenvalue index corresponds to the eigenvalues (frequencies) of the graph Laplacian $\boldsymbol{L}$. The convoluted single-channel signal in the Fourier domain is the element-wise scalar product of the filter and signal.

of this signal for each output feature channel. While the exact formulation is computationally prohibitive for large graphs, knowing the exact form of the graph convolution for multiple channels will allow for various approximations. We define required properties for localized approximations that similarly operate on multiple sparse edge relations. We further introduce the multi-channel graph isomorphism network (MC-GIN) that we prove to have the same expressivity as the graph isomorphism network (Xu et al., 2019) while applying a linear transformation to the data. Our experiments confirm the advantages of the MCGC and MC-GIN. We summarize our main contributions as follows:

- Based on the general definition of the convolution for multi-channel signals, we obtain the multi-channel graph convolution (MCGCs) using the convolution theorem (Section 3).
- To construct multi-channel MPNNs (MC-MPNNs), we introduce localized MCGCs, for which we require multiple edge relations, with each edge relation amplifying a different signal in the data (Section 3.1).
- We prove that with localized MCGCs, we can construct linear mappings that are injective on multisets, which we call multi-channel graph isomorphism network (MC-GIN) due to its expressivity being equivalent to the GIN (Xu et al., 2019) (Section 3.2).

## 2 PRELIMINARIES

Let $\mathcal{G} = (\mathcal{V}, \mathcal{E})$ be a connected and undirected graph consisting of a set of $n$ nodes $\mathcal{V}$ and a set of edges $\mathcal{E}$. A graph signal is defined as a function $x \colon \mathcal{V} \to \mathbb{R}^d$ that assigns a vector of real values to each node. For notational simplicity, we stack all node signals into a matrix $\boldsymbol{X} \in \mathbb{R}^{n \times d}$ based on some node ordering. For $d = 1$, we refer to this as a single-channel signal, while for $d > 1$, we call it a multi-channel signal. These can either be initial features or expressive and informative node embeddings obtained by a suitable method. Let $\boldsymbol{A} \in \{0, 1\}^{n \times n}$ with $A_{i,j} = 1$ if $(i, j) \in \mathcal{E}$ and $0$ otherwise be the adjacency matrix corresponding to the same node ordering as $\boldsymbol{X}$. The diagonal degree matrix is $\boldsymbol{D} \in \mathbb{N}^{n \times n}$. The symmetrically normalized adjacency matrix is given by $\tilde{\boldsymbol{A}} = \boldsymbol{D}^{-1/2} \boldsymbol{A} \boldsymbol{D}^{-1/2}$ and the graph Laplacian by $\boldsymbol{L} = \boldsymbol{I}_n - \tilde{\boldsymbol{A}}$. Its eigendecomposition is $\boldsymbol{L} = \boldsymbol{U} \boldsymbol{\Lambda} \boldsymbol{U}^T$ where $\boldsymbol{\Lambda} \in \mathbb{R}^{n \times n}$ is a diagonal matrix containing its eigenvalues, and $\boldsymbol{U} \in \mathbb{R}^{n \times n}$ is an orthonormal matrix containing the corresponding eigenvectors as columns. In the graph domain, the Fourier base is given by the eigenvectors $\boldsymbol{U}^T$ of the graph Laplacian. Thus, the Fourier transformation $F = \boldsymbol{U}^T$ is performed by projecting a graph signal onto the eigenvectors, and its inverse transformation is given by $F^{-1} = \boldsymbol{U}$.

### 2.1 SINGLE-CHANNEL GRAPH CONVOLUTIONS

The convolution is an operation that combines two functions and produces a new function. Given a graph with $n$ nodes, the convolution is typically defined as the function $\boldsymbol{x}' = \boldsymbol{w} * \boldsymbol{x}$, where $\boldsymbol{x} \in \mathbb{R}^n$ is a single-channel graph signal, $\boldsymbol{w} \in \mathbb{R}^n$ is a corresponding filter, and it produces a convoluted single-channel graph signal $\boldsymbol{w}' \in \mathbb{R}^{n}$. As this convolution involves single-channel signals, we will further

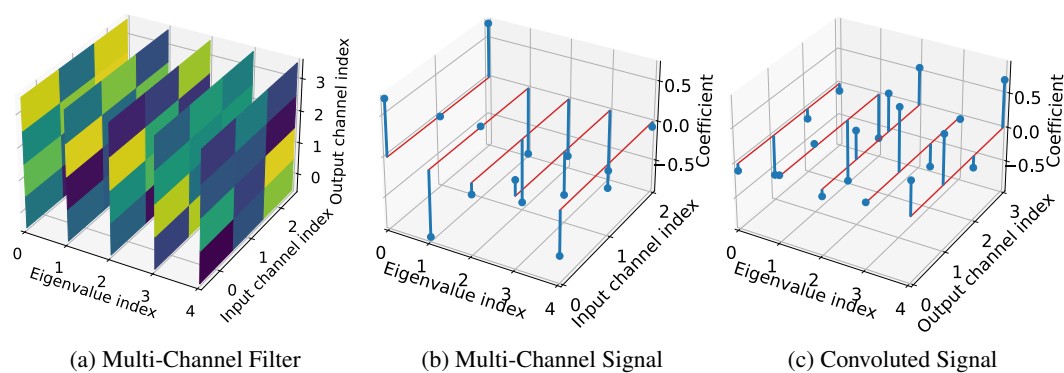

(a) Multi-Channel Filter    (b) Multi-Channel Signal    (c) Convoluted Signal

Figure 2: Multi-channel signal and filter in the Fourier domain. The eigenvalue index corresponds to the eigenvalues (frequencies) of the graph Laplacian $\boldsymbol{L}$. The element-wise product for each eigenvalue index is a matrix-vector product.

refer to it as the single-channel graph convolution (SCGC). The convolution theorem (O'Neil, 1963) states that the Fourier transform of a convolution

$$F(\boldsymbol{w} * \boldsymbol{x}) = F(\boldsymbol{w}) \odot F(\boldsymbol{x})$$

is equal to the element-wise multiplication of the filter and signal in the Fourier domain. The graph convolution (Hammond et al., 2011) can be expressed as

$$\boldsymbol{w} * \boldsymbol{x} = F^{-1}(F(\boldsymbol{w}) \odot F(\boldsymbol{x}))$$
$$= \boldsymbol{U}(\boldsymbol{U}^T\boldsymbol{w} \odot \boldsymbol{U}^T\boldsymbol{x})$$

by utilizing the graph Fourier transform $F = \boldsymbol{U}^T$ and its inverse $F^{-1} = \boldsymbol{U}$ based on the eigenvectors $\boldsymbol{U}$ of the graph Laplacian $\boldsymbol{L}$. We visualize this process in Figure 1. Substituting $\boldsymbol{U}^T\boldsymbol{w}$ with its diagonalized matrix $\boldsymbol{W}^* = \mathrm{diag}(\boldsymbol{U}^T\boldsymbol{w})$ and the Hadamard product with a matrix multiplication leads to the equivalent form

$$\boldsymbol{U}(\boldsymbol{U}^T\boldsymbol{w} \odot \boldsymbol{U}^T\boldsymbol{x}) = \boldsymbol{U}\boldsymbol{W}^*\boldsymbol{U}^T\boldsymbol{x}\,.$$

Bruna et al. (2014) proposed to learn the filter $\boldsymbol{W}^*$ directly in the Fourier domain, which is also known as the spectral graph convolution.

## 2.2 Approximations using Polynomials

As computing the eigendecomposition and performing dense matrix multiplications is computationally expensive, SCGCs do not scale well to large graphs. Defferrard et al. (2016) proposed to instead approximate and parameterize graph convolutions using a learnable function $g(\boldsymbol{\Lambda})$ on the eigenvalues $\boldsymbol{\Lambda}$ of $\boldsymbol{L}$, i.e., by approximating the diagonal matrix $\boldsymbol{W}^* \approx g(\boldsymbol{\Lambda})$. One commonly employed approximation

$$\boldsymbol{U}\boldsymbol{W}^*\boldsymbol{U}^T \approx \boldsymbol{U}\sum_{p=1}^{n} w_p \boldsymbol{U}T_p(\tilde{\boldsymbol{\Lambda}})\boldsymbol{U}^T$$

$$= \sum_{p=0}^{k} w_p T_p(\boldsymbol{U}\boldsymbol{\Lambda}^*\boldsymbol{U}^T)$$

$$= \sum_{p=0}^{k} w_p T_p(\tilde{\boldsymbol{L}})$$

is based on Chebyshev polynomials (Chebyshev, 1853) up to $k$-th order of a rescaled matrix of eigenvectors $\tilde{\boldsymbol{\Lambda}} = \frac{2}{\lambda_{\max}}\boldsymbol{\Lambda} - \boldsymbol{I}_n$ and corresponding $\tilde{\boldsymbol{L}} = \frac{1}{\lambda_{\max}} - \boldsymbol{I}_n$ (Hammond et al., 2011). The polynomials are defined as $T_0(\boldsymbol{\Lambda}) = \boldsymbol{I}$, $T_1(\boldsymbol{\Lambda}) = \boldsymbol{\Lambda}$, and $T_k(\boldsymbol{\Lambda}) = 2\boldsymbol{\Lambda}T_{k-1}(\boldsymbol{\Lambda}) - T_{k-2}(\boldsymbol{\Lambda})$. This

truncated polynomial expansion is $k$-localized in the graph as entries $\left[\sum_{p=0}^{k} w_p T_p(\tilde{L})\right]_{i,j}$ are zero when the shortest path between nodes $i$ and $j$ is larger than $k$. Further approximations include Cayley polynomials (Levie et al., 2019), Bernstein polynomials (He et al., 2021). Koke & Cremers (2024) propose an approximation for directed graphs using Faber polynomials. GPR-GNN proposes to directly learn the filter weights (Chien et al., 2021). All these methods are approximations of the SGCG.

### 2.3 MESSAGE-PASSING NEURAL NETWORKS

For further computational efficiency and empirical success, the graph convolutional network (GCN) (Kipf & Welling, 2017) was derived as a 1-localized approximation of Chebyshev polynomials. It uses the approximation

$$\boldsymbol{w} * \boldsymbol{x} \approx w_0 (\boldsymbol{I}_n + \boldsymbol{D}^{-1/2} \boldsymbol{A} \boldsymbol{D}^{-1/2}) \boldsymbol{x}.$$

by setting $k = 1$, $\lambda_{\max} \approx 2$, and $w_0 = -w_1$. They further substitute $\boldsymbol{I}_n + \boldsymbol{D}^{-1/2} \boldsymbol{A} \boldsymbol{D}^{-1/2}$ by $\tilde{\boldsymbol{D}}^{-1/2} \tilde{\boldsymbol{A}} \tilde{\boldsymbol{D}}^{-1/2}$ using $\tilde{\boldsymbol{A}} = \boldsymbol{A} + \boldsymbol{I}_n$ and $\tilde{\boldsymbol{D}} = \boldsymbol{D} + \boldsymbol{I}_n$.

However, as the SCGC is not defined for multi-channel signals, the SCGC and its approximations are not directly applicable to multi-channel signals. To apply the GCN to a multi-channel signal $\boldsymbol{X} \in \mathbb{R}^{n \times d}$ or to obtain a multi-channel signal $\boldsymbol{X}' \in \mathbb{R}^{n \times d'}$ as the output, they propose to replace the scalar $w \in \mathbb{R}$ with a matrix $\boldsymbol{W} \in \mathbb{R}^{d \times d'}$ (Kipf & Welling, 2017). This leads to their update function

$$\boldsymbol{X}' = \hat{\boldsymbol{A}} \boldsymbol{X} \boldsymbol{W}, \tag{1}$$

where $\hat{\boldsymbol{A}} = \tilde{\boldsymbol{D}}^{-1/2} \tilde{\boldsymbol{A}} \tilde{\boldsymbol{D}}^{-1/2}$.

Recent research on over-smoothing and rank collapse has proven that graph convolutions of the form in Eq. 1 do not allow the amplification of different signals across channels (Giovanni et al., 2023; Roth & Liebig, 2023; Roth, 2024). The signals to be amplified are solely determined by the spectrum of $\hat{\boldsymbol{A}}$, which was shown to hold for any choice of $\hat{\boldsymbol{A}}$ and $\boldsymbol{W}$. While this may be desired if the same signal should be amplified for all feature columns, each feature channel should typically be composed of a different mix of signals. Thus, the GCN inherited this issue from SCGCs.

Most established message-passing neural networks (MPNNs) are adaptations of the GCN and similarly suffer from being based on single-channel convolutions. Examples include utilizing the mean (Hamilton et al., 2017) or sum (Xu et al., 2019) aggregation, using the attention mechanism (Velickovic et al., 2018; Brody et al., 2022), allowing for negative edge weights (Yan et al., 2022) or building on top of these graph convolutions with normalization layers (Zhao & Akoglu, 2020), gating mechanisms (Li et al., 2016; Rusch et al., 2023), or positional encodings (Kreuzer et al., 2021; Rampásek et al., 2022; Huang et al., 2024). We will thus consider how to directly define the graph convolution for multi-channel signals, of which all approximations would benefit.

## 3 MULTI-CHANNEL GRAPH CONVOLUTIONS

We now consider a multi-channel graph signal $\boldsymbol{X} \in \mathbb{R}^{n \times d}$ with $d$ channels for each node. By applying a graph convolution, we want to obtain a convoluted multi-channel graph signal with different channel mixing for any input and output combination. In the last section, we have shown that the currently used SCGC is not defined for this task. In general, the convolution is defined for multi-channel signals $\boldsymbol{X}$. To obtain a convoluted signal with $c$ channels, the element-wise product needs to map vectors with $d$ channels to vectors with $c$ channels. This requires elements of the filter to be matrices $\mathsf{W}_i \in \mathbb{R}^{c \times d}$ and the full filter to be $\mathsf{W} \in \mathbb{R}^{n \times c \times d}$. This is commonly referred to as the multi-channel convolution Burg (1964); Inouye & Sato (1999). It is commonly used in signal processing (Burg, 1964; Inouye & Sato, 1999; Sainath et al., 2017), e.g., for multi-channel Wiener filtering (Brandstein & Ward, 2001) and finite impulse responses (Seltzer et al., 2004). Convolutional neural networks (CNNs) (LeCun et al., 1989) are similarly based on the multi-channel convolution (Zhang et al., 2021).

Equivalently to the single-channel convolution, we can compute the multi-channel graph convolution in the Fourier domain using the convolutional theorem. The exact form of the multi-channel

convolution on graphs is a direct consequence of the general multi-channel convolution and the graph Fourier transform:

**Theorem 3.1.** *(Multi-Channel Graph Convolution) Let $\boldsymbol{L} \in \mathbb{R}^{n \times n}$ be a graph Laplacian of an undirected graph and the graph Fourier transform be given by its matrix of eigenvectors $\boldsymbol{U}^T \in \mathbb{R}^{n \times n}$. Further, let $\boldsymbol{X} \in \mathbb{R}^{n \times d}$ be a multi-channel graph signal, and $\mathbf{W} \in \mathbb{R}^{n \times c \times d}$ be a corresponding filter matrix. Then,*

$$\mathbf{W} * \boldsymbol{X} = \sum_{i=1}^{n} \boldsymbol{u}_i \boldsymbol{u}_i^T \boldsymbol{X} \hat{\boldsymbol{W}}_i \tag{2}$$

*where $\hat{\boldsymbol{W}}_k = [\boldsymbol{U}^T \mathbf{W}]_k^T$ is the graph Fourier transformed filter, $\boldsymbol{u}_k$ is the $k$-th eigenvector of $\boldsymbol{L}$.*

*Proof.* We use the vectorized signal $\hat{\boldsymbol{x}} = \text{vec}(\boldsymbol{X}) \in \mathbb{R}^{n \cdot c}$ by stacking its columns. The graph Fourier transform on matrices and tensors is applied along the node dimension, i.e., independently on each channel. For matrix $\boldsymbol{X}$ this results in

$$F(\boldsymbol{X}) = \boldsymbol{U}^T \boldsymbol{X} \in \mathbb{R}^{n \times d}$$

and for tensor $\mathbf{W}$ in

$$\hat{\mathbf{W}} = F(\mathbf{W}) = \boldsymbol{U}^T \times_1 \mathbf{W} \in \mathbb{R}^{n \times c \times d}$$

where $\times_1$ is the 1-mode tensor matrix product (Kolda & Bader, 2009) that performs the desired broadcasted matrix multiplication. With this, we state the multi-channel graph convolution in the Fourier domain as

$$\mathbf{W} * \boldsymbol{X} = \boldsymbol{U}(\boldsymbol{U}^T \times_1 \mathbf{W} \odot \boldsymbol{U}^T \boldsymbol{X}) = \boldsymbol{U}(\hat{\mathbf{W}} \odot \boldsymbol{U}^T \boldsymbol{X}).$$

This product is visualized in Figure 2. Similarly to the single-channel, we simplify this expression using matrix multiplications. Equivalently to the single-channel case, this can be achieved by diagonalizing $\hat{\mathbf{W}}$ into a block matrix of diagonal blocks

$$\boldsymbol{D} = \begin{bmatrix} \hat{W}_{1,0,0} & 0 & 0 & & \hat{W}_{1,0,d} & 0 & 0 \\ 0 & \ddots & 0 & \dots & 0 & \ddots & 0 \\ 0 & 0 & \hat{W}_{n,0,0} & & 0 & 0 & \hat{W}_{n,0,d} \\ & \vdots & & \ddots & & \vdots & \\ \hat{W}_{1,c,0} & 0 & 0 & & \hat{W}_{1,c,d} & 0 & 0 \\ 0 & \ddots & 0 & \dots & 0 & \ddots & 0 \\ 0 & 0 & \hat{W}_{n,c,0} & & 0 & 0 & \hat{W}_{n,c,d} \end{bmatrix} \in \mathbb{R}^{nc \times nd}$$

where each $\hat{W}_{k,i,j} \in \mathbb{R}$ corresponds to position $i, j$ in $\hat{\mathbf{W}}_k \in \mathbb{R}^{c \times d}$. This simplifies the equivalent vectorized form into

$$\text{vec}(\hat{\mathbf{W}} \odot \boldsymbol{U}^T \boldsymbol{X}) = \boldsymbol{D}\text{vec}(\boldsymbol{U}^T \boldsymbol{X}) = \boldsymbol{D}\left(\boldsymbol{I}_d \otimes \boldsymbol{U}^T\right) \text{vec}(\boldsymbol{X})$$

by utilizing the Kronecker product $\otimes$. The matrix $\boldsymbol{D}$ can further be decomposed into a sum of Kronecker products $\boldsymbol{D} = \sum_{i=1}^{n} \hat{\mathbf{W}}_i \otimes \boldsymbol{I}_n^{(i)}$, where for each $\boldsymbol{I}_n^{(i)} \in \mathbb{R}^{n \times n}$ all entries are zero, apart from position $i, i$ which is one. This lets us state the full vectorized multi-channel graph convolution as

$$\text{vec}(\mathbf{W} * \boldsymbol{X}) = (\boldsymbol{I}_n \otimes \boldsymbol{U}) \left( \sum_{i=1}^{n} \hat{\mathbf{W}}_i \otimes \boldsymbol{I}_n^{(i)} \right) \left( \boldsymbol{I}_n \otimes \boldsymbol{U}^T \right) \text{vec}(\boldsymbol{X})$$

$$= \left( \sum_{i=1}^{n} \hat{\mathbf{W}}_i \otimes \boldsymbol{u}_i \boldsymbol{u}_i^T \right) \text{vec}(\boldsymbol{X})$$

by using the fact that $\boldsymbol{U}\boldsymbol{I}_n^{(i)}\boldsymbol{U}^T = \boldsymbol{u}_i \boldsymbol{u}_i^T$. Inverting the vec operation allows us to avoid the Kronecker product and state the exact multi-channel graph convolution as

$$\boldsymbol{W} * \boldsymbol{X} = \sum_{i=1}^{n} \boldsymbol{u}_i \boldsymbol{u}_i^T \boldsymbol{X} \hat{\mathbf{W}}_i^T$$

This concludes the proof. □

We emphasize that this form is a mathematical fact and not a definition made by us. We interpret the multi-channel graph convolutions as follows: Each term $\boldsymbol{u}_i \boldsymbol{u}_i^T \boldsymbol{X} \hat{\boldsymbol{W}}_i$ corresponds to one Fourier basis vector. The corresponding parameter matrix $\hat{\boldsymbol{W}}_i$ specifies how much this signal coming from each input channel should be amplified or damped for each output channel. Utilizing $n$ terms allows the multi-channel graph convolution to amplify and filter specific input signals for each output channel.

Now that we have obtained the graph convolution for multi-channel input or output signals, we could directly apply it to suitable tasks. However, we expect similar issues for the multi-channel graph convolution as we have seen for the single-channel graph convolution. Issues with directly computing the single-channel graph convolution include high computational costs due to dense matrix multiplication and computing the eigendecomposition. It is also inherently transductive, as the eigenvectors change across graphs, and thus, a learned filter is not applicable. It was also observed that the single-channel graph convolution quickly overfits given training data. Many directions have been proposed to approximate the single-channel convolution, as outlined in Section 2.1. The most prominent approximation and derivation from single-channel graph convolutions are MPNNs. While a similar multitude of novel approximations can be constructed for the multi-channel graph convolution, we want to outline how MPNNs based on MCGC should be designed.

## 3.1 MULTI-CHANNEL MPNNS

Each outer product $\boldsymbol{u}_i \boldsymbol{u}_i^T$ can be seen as one edge relation type with full connectivity and edge weights $u_{i,p} u_{i,q}$ between nodes $p$ and $q$. Multi-channel MPNNs should be localized and still utilize multiple edge relations $\hat{\boldsymbol{A}}^{(1)}, \ldots, \hat{\boldsymbol{A}}^{(k)} \in \mathbb{R}^{n \times n}$, with each relation amplifying a different signal. While a sparse graph will have more than one non-zero eigenvalue, its dominating eigenvector can be controlled by its edge weights. The corresponding transformations $\boldsymbol{W}^{(1)}, \ldots, \boldsymbol{W}^{(k)} \in \mathbb{R}^{d \times d'}$ can then similarly determine the amplification or damping of these signals for each channel. Splitting a graph into a multi-relational graph was recently proposed to avoid representational rank collapse (Roth et al., 2024). We define localized multi-channel graph convolutions as follows:

**Definition 3.2.** (Localized Multi-Channel Graph Convolution (l-MCGC)) Let $\boldsymbol{A} \in \mathbb{R}^{n \times n}$ be an adjacency matrix. A function

$$\phi(\boldsymbol{X}) = \boldsymbol{X}' = \sum_{m=1}^{k} \hat{\boldsymbol{A}}^{(m)} \boldsymbol{X} \boldsymbol{W}^{(m)} \tag{3}$$

with $\boldsymbol{X} \in \mathbb{R}^{n \times d}$, $\hat{\boldsymbol{A}}^{(1)}, \ldots, \hat{\boldsymbol{A}}^{(k)} \in \mathbb{R}^{n \times n}$, and $\boldsymbol{W}^{(1)}, \ldots, \boldsymbol{W}^{(k)} \in \mathbb{R}^{d \times c}$ is called a localized multi-channel graph convolution (l-MCGC) if the following two conditions are satisfied:

- the dominant eigenvectors of $\hat{\boldsymbol{A}}^{(1)}, \ldots, \hat{\boldsymbol{A}}^{(k)}$ are linearly independent and

- $\forall i, j \in \{1, \ldots, n\}, l \in \{1, \ldots, k\} : \boldsymbol{A}_{ij} = 0 \implies \hat{\boldsymbol{A}}_{ij}^{(l)} = 0.$

These l-MCGCs can be equivalently expressed as a node-based update function

$$\boldsymbol{x}_i' = \phi(\boldsymbol{X})_i = \sum_{m=1}^{k} \sum_{j \in \mathbb{N}_i} \hat{a}_{i,j}^{(m)} \boldsymbol{x}_j \boldsymbol{W}^{(m)}$$

$$= \sum_{j \in \mathbb{N}_i} \boldsymbol{x}_j \boldsymbol{W}^{(i,j)}$$

for node $i$ using its set of neighbors $\mathbb{N}_i$, edge weights $a_{i,j}^{(m)} = A_{i,j}^{(m)}$. For each message, a linear combination $\boldsymbol{W}^{(i,j)} = \sum_{m=1}^{k} \hat{a}_{i,j}^{(m)} \boldsymbol{W}^{(m)}$ of feature transformations $\boldsymbol{W}^{(1)}, \ldots, \boldsymbol{W}^{(k)}$ is applied to the corresponding node state. This form also confirms that l-MCGCs remain permutation equivariant when requiring all $\hat{a}_{i,j}^{(m)}$ to be obtained using a permutation equivariant method. There are many ways to obtain a set of graphs $\hat{\boldsymbol{A}}^{(1)}, \ldots, \hat{\boldsymbol{A}}^{(k)}$ with distinct eigenvectors. As graph attention networks (Velickovic et al., 2018) with multiple attention heads can be expressed in this form as well, we will similarly refer to each term as a head. However, each $\boldsymbol{A}^{(1)}, \ldots, \boldsymbol{A}^{(k)}$ is a row-stochastic

matrix because of the softmax activation. As all row-stochastic matrices have the constant vector as dominant eigenvector (Asmussen, 2003), all heads amplify the same signal. We now identify a connection between l-MCGCs and their expressivity with respect to the Weisfeiler-Leman test (Weisfeiler & Lehman, 1968), which leads to one possible instantiation of l-MCGCs.

## 3.2 EXPRESSIVITY

Our analysis has shown the advantages of multi-channel graph convolutions for amplifying different signals across channels. However, much attention has been paid to the structural expressivity of MPNNs, which studies the ability of MPNNs to distinguish non-isomorphic graphs. It was determined that MPNNs are upper-bounded in expressivity by the Weisfeiler-Leman test (Xu et al., 2019; Morris et al., 2019). To achieve this maximal expressivity, the graph isomorphism network (GIN) Xu et al. (2019) is typically employed and applies a complex non-linear feature transformation. We show that we can achieve the same expressivity by obtaining a linear l-MCGC:

**Proposition 3.3.** *(Injectivity on multisets.) Let $\mathbb{X} = \{\{\boldsymbol{x}_1, \ldots, \boldsymbol{x}_n\}\}$ be a countable multiset with $\boldsymbol{x}_i \in \mathbb{R}^{1 \times d}$ for $d \in \mathbb{N}$. Then, there exists a function $f \colon \mathbb{R}^d \times \mathbb{R}^d \to \mathbb{R}^k$ so that*

$$\boldsymbol{x}_i' = \sum_{l=1}^{k} \sum_{\boldsymbol{x}_j \in \mathbb{X}_i} f(\boldsymbol{x}_i, \boldsymbol{x}_j)_l \boldsymbol{x}_j \boldsymbol{W}^{(l)}$$

*is injective for all $\boldsymbol{x}_i \in \mathbb{X}$ and $\mathbb{X}_i \subset \mathbb{X}$ of bounded size, all $k > 1$, and a.e. $\boldsymbol{W}^{(1)}, \ldots, \boldsymbol{W}^{(k)} \in \mathbb{R}^{d \times d'}$ with $d' \in \{1, \ldots, n\}$.*

*Proof.* Given any $\boldsymbol{x}_j \in \mathbb{X}$, two linear transformations $\boldsymbol{W}^{(m)}, \ldots, \boldsymbol{W}^{(n)}$ map $\boldsymbol{x}_j$ to pairwise linearly independent vectors $\boldsymbol{y}_j^{(m)} = \boldsymbol{x}_j \boldsymbol{W}^{(m)}$ and $\boldsymbol{y}_j^{(n)} = \boldsymbol{x}_j \boldsymbol{W}^{(n)}$ for a.e. $\boldsymbol{W}^{(m)}, \ldots, \boldsymbol{W}^{(n)}$ with respect to the Lebesgue measure.

To prove the existence of the desired $f$, we follow the proof of Lemma 5 in Xu et al. (2019) (injectivity of GIN): Because $\mathbb{X}$ is countable, there exist injective mappings $Z_1 \colon \mathbb{X} \to \mathbb{N}$, $Z_2 \colon \mathbb{X} \to \mathbb{N}$ mapping elements $\boldsymbol{x}_m \in \mathbb{X}$ to natural numbers. As each $\mathbb{X}_n$ is of bounded size, there exists a number $N \in \mathbb{N}$ so that $|\mathbb{X}| < N$ for all $X$. An example of such $f$ is $f(\boldsymbol{x}_m, \boldsymbol{x}_n) = \begin{bmatrix} N^{-Z_1(\boldsymbol{x}_m)} & 1 & \ldots & 1 & N^{-Z_2(\boldsymbol{x}_n)} \end{bmatrix}$. The first and last output values of $f$ can be viewed as continuous one-hot vectors of $\boldsymbol{x}_m$ and $\boldsymbol{x}_n$, respectively. Because $\mathbb{X}$ is countable, $Z_1$ and $Z_2$ can further be chosen to output linearly independent vectors for all pairs of distinct inputs.

Thus, for all $\boldsymbol{x}_i \in \mathbb{X}$ and $\boldsymbol{x}_j \in \mathbb{X}$, we have a different linear combination of pairwise linear independent vectors and pairwise linearly independent linear combinations, and thus $\boldsymbol{s}_{i,j} = \sum_{l=1}^{k} f(\boldsymbol{x}_i, \boldsymbol{x}_j)_l \boldsymbol{y}_j^{(l)}$ is pairwise linearly independent for all $i, j$, and a.e. choice of $\boldsymbol{W}^{(1)}, \ldots, \boldsymbol{W}^{(k)}$.

Each node state is updated as the sum

$$\boldsymbol{x}_i' = \sum_{\boldsymbol{x}_j \in \mathbb{X}_i} \boldsymbol{s}_{i,j}$$

of these terms, which is also linearly independent for a.e. choice of $\boldsymbol{W}^{(1)}, \ldots, \boldsymbol{W}^{(k)}$. Thus, all $\boldsymbol{x}_i'$ are linearly independent for different $\boldsymbol{x}_i \in \mathbb{X}$ and $\mathbb{X}_i \subset \mathbb{X}$ which implies injectivity. □

By this injectivity, the expressivity of l-MCGCs is equivalent to that of the graph isomorphism network (GIN) (Xu et al., 2019), and thus also to the Weisfeiler-Leman test (Weisfeiler & Lehman, 1968). This is the maximal achievable expressivity for models following the message-passing scheme. Due to the universal approximation theorem (Hornik et al., 1989; Hornik, 1991), $f$ can be represented by a multi-layer perception (MLP). GIN similarly utilizes an injective function $f$ that is modeled by an MLP, which is applied instead of the linear feature transformation $\boldsymbol{W}_1$ with $k = 1$. We will further to this l-MCGC as the multi-channel graph isomorphism network (MC-GIN). This is a critical advantage of MC-GINs: As convolutions are linear operators, it is desired to approximate multi-channel graph convolutions similarly by a linear transformation on $\boldsymbol{X}$. While $f$ may introduce a complex transformation, it constructs a linear transformation that is applied to $\boldsymbol{X}$. Allowing for learnable edge weights also lifts the constraint of requiring the sum as aggregation.

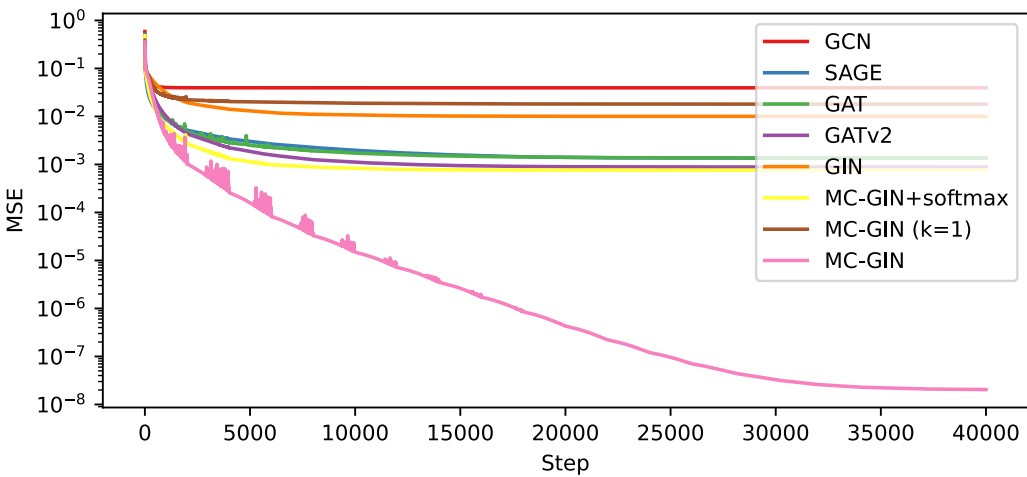

Figure 3: Mean squared error (MSE) during optimization of our function approximation task by applying a single-layer of different methods.

## 4 EXPERIMENTS

Our aim with this work is to introduce MCGCs in general and to inspire future research to find efficient and effective approximations that will lead to state-of-the-art results. We briefly want to empirically confirm the potential of MCGCs for learning on graphs using one synthetic and one benchmark task. Additional details on models, datasets, and hyperparameters can be found in Appendix A.3. Our implementation is based on PyTorch Geometric (Fey & Lenssen, 2019).

### 4.1 FUNCTION APPROXIMATION

We first want to evaluate whether the MCGC and the MC-GIN can obtain more informative embeddings by approximating the mapping $Y = \phi(X)$ where $X, Y \in \mathbb{R}^{n \times d}$ are multi-channel signals of random but fixed values $X_{i,j} \sim \mathcal{N}(0, 1)$ and $Y_{i,j} \sim \mathcal{N}(0, 1)$. For $\phi$, we evaluate a single layer of several graph convolutional operators. Namely the GCN (Kipf & Welling, 2017), GAT (Velickovic et al., 2018), GATv2 (Brody et al., 2022), GIN (Xu et al., 2019), the SAGE convolution (Hamilton et al., 2017), our proposed MC-GIN. We set the number of heads for GAT, GATv2, and MC-GIN to $k = 2$. For functions $f$ in GIN and MC-GIN, we utilize a three-layer MLP with ReLU activations after the first two layers. As an ablation, we further evaluate two versions that do not satisfy our required property for multi-channel MPNNs. We apply a softmax activation to incoming edge weights obtained by $f$ for each node and head with $k = 2$, equivalently to attention (MC-GIN+softmax), and MC-GIN with $k = 1$. We also evaluate the MC-GIN with $k = 3$ and MC-GIN with a one-layer MLP for $f$ and MC-GIN with a two-layer MLP, which satisfy our multi-channel properties. We additionally present results for the MCGC, which, in theory, can represent any such mapping exactly. As the benefits of MCGCs do not depend on specific graph properties, we sample a random undirected Erdős–Rényi graph (Erdős & Rényi, 1959) with $n$ nodes and an edge probability of $5\%$. We minimize the mean-squared error (MSE) between $\phi(X)$ and $Y$ using the Adam optimizer. The base learning rate is tuned in $\{0.01, 0.003, 0.001, 0.0003\}$ for each method and is halved every 2000 steps. We set $n = 64$ and $d = 32$.

The averaged loss progression over five runs during optimization is visualized in Figure 3, and the average minimum MSE scores after $40\,000$ steps are presented in Table 1. MC-GIN achieves an improved approximation error by at least four orders of magnitude compared to all other methods. Notably, all methods that use a single head (GCN, GIN, and MC-GIN (k=1)) achieve the worst approximation. Similarly, the error for methods that apply the softmax activation (GAT, GATv2, and MC-GIN+softmax) is significantly higher, as each head amplifies the same signal for all channels. With our ablations, we also find the choice for $f$ not to influence the results strongly (MC-GIN 1 layer MLP, MC-GIN 2 layer MLP) and an additional head to further improve the results (MC-GIN

| Method | MSE |
|---|---|
| GCN | $39 \cdot 10^{-3} \pm 10 \cdot 10^{-4}$ |
| GAT | $14 \cdot 10^{-4} \pm 12 \cdot 10^{-4}$ |
| GATv2 | $\underline{88 \cdot 10^{-5}} \pm 15 \cdot 10^{-4}$ |
| SAGE | $14 \cdot 10^{-4} \pm 46 \cdot 10^{-5}$ |
| GIN | $10 \cdot 10^{-3} \pm 35 \cdot 10^{-4}$ |
| MC-GIN | $\mathbf{20 \cdot 10^{-9}} \pm 41 \cdot 10^{-9}$ |
| MC-GIN+softmax | $76 \cdot 10^{-5} \pm 15 \cdot 10^{-4}$ |
| MC-GIN (k=1) | $18 \cdot 10^{-3} \pm 27 \cdot 10^{-4}$ |
| MC-GIN (k=3) | $20 \cdot 10^{-15} \pm 17 \cdot 10^{-15}$ |
| MC-GIN (1 layer MLP) | $18 \cdot 10^{-11} \pm 23 \cdot 10^{-11}$ |
| MC-GIN (2 layer MLP) | $15 \cdot 10^{-9} \pm 17 \cdot 10^{-9}$ |
| MCGC | $37 \cdot 10^{-16} \pm 44 \cdot 10^{-17}$ |

Table 1: Average and standard deviation of the minimal mean-squared error (MSE). Best MSE in **bold**, second-best underlined.

| Method | MAE | | Time |
|---|---|---|---|
| | Train | Test | (s) |
| GCN | $6.2 \pm 0.2$ | $16.0 \pm 0.2$ | **32** |
| GATv2 | $5.7 \pm 0.2$ | $13.6 \pm 0.6$ | 63 |
| SAGE | $\underline{3.9} \pm 0.1$ | $12.3 \pm 0.2$ | $\underline{36}$ |
| GIN | $5.8 \pm 0.1$ | $\underline{12.3} \pm 0.4$ | 42 |
| MC-GIN | $\mathbf{3.3} \pm 0.2$ | $\mathbf{10.5} \pm 0.2$ | 52 |

Table 2: Results on ZINC. Mean absolute error (MAE) scores are multiplied by 100 for clarity. Best scores in **bold**, second-best underlined.

(k=3). While these improved capabilities come with the risk of overfitting noise in the data, MCGCs are beneficial for complex tasks in which MPNNs struggle to achieve satisfying performance.

## 4.2 ZINC

We now consider the ZINC dataset (Sterling & Irwin, 2015) to show that approximations of the MCGC can generally also improve benchmark results. It consists of around $250\,000$ molecular graphs, with the task being to predict the constrained solubility of each molecule. We integrate several base message-passing layers (GCN (Kipf & Welling, 2017), Gatv2 (Brody et al., 2022), SAGE (Hamilton et al., 2017), GIN (Xu et al., 2019), and MC-GIN with $k = 2$) into the implementation of the Long Range Graph Benchmark (Dwivedi et al., 2022) and the updated training scheme of Tönshoff et al. (2024). The number of layers and the learning rate are tuned for each method. As proposed by Dwivedi et al. (2022), each model utilizes at most $500\,000$ parameters.

Average train and test errors with independent optimal hyperparameters are presented in Table 2. MC-GIN improves both train and test loss by at least $15\%$ compared to the four other methods. The execution time of our implementation is increased by around $24\%$ compared to the GIN. While countless combinations with other techniques, datasets, and tasks can be evaluated, we want to motivate general research on approximating MCGC that will eventually lead to state-of-the-art results.

## 5 Conclusion

In this work, we introduced multi-channel convolutions to graphs by obtaining their form based on the convolution theorem and the graph Fourier transform. While the currently used single-channel graph convolutions, their polynomial approximations, and most MPNNs are not defined for nodes with multiple features, MCGCs are specifically defined for multi-channel signals. This allows MCGCs to obtain more informative embeddings, as different signals can be amplified for each feature channel. Corresponding multi-channel MPNNs need to utilize multiple aggregations and transformations, each amplifying a different signal. We introduce the multi-channel graph isomorphism network (MC-GIN), which can obtain a linear mapping with equivalent expressive power to the graph isomorphism network (Xu et al., 2019). Our experiments confirm the strongly improved abilities to fit complex functions. While our experimental evaluation is limited, the benefits of MCGCs compared to SCGCs are very clear. Having access to the mathematically correct convolution for multi-channel signals allows for the development of various approximations. As more complex interactions between channels may not be required for all tasks, MCGCs increase the risk of overfitting. Approximations of the MCGC will be particularly important as more complex graph-related tasks emerge and more powerful convolutions are needed.

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

## A    EXPERIMENTAL DETAILS

Experiments on ZINC were executed on an Nvidia H100 GPU with 96 GB, and experiments on function approximation were executed on an Intel Xeon 8468 Sapphire with 48 cores.

### A.1    METHODS

For all methods, we use their standard implementation from PyTorch Geometric (Fey & Lenssen, 2019). We mostly use their default parameters but do not add self-loops to any method. For GAT and GATv2, we use two heads and compute the element-wise mean over the outputs of both heads. For GIN, the MLP consists of three linear layers with a bias, followed by a ReLU activation after the first two layers. For MC-GIN, the two adjacent node states of each edge are concatenated, and the same three-layer MLP is applied. Only the number of input channels doubles and the number of output channels is set to the number of heads. The number of heads is set to $k = 2$ unless explicitly stated otherwise.

### A.2    ZINC

The ZINC dataset (Sterling & Irwin, 2015) consists of $249\,456$ graphs, each representing a molecule. Nodes in the graphs represent to heavy atoms, and edges correspond to their connectivity. Each molecule consists of $23.2$ nodes and $49.8$ edges on average. The task is to predict the constrained solubility, which is a graph regression task. The type of atom is the only feature given. A batch size of 32 is used. The mean absolute error (MAE) is used for optimization and reporting on the test set. Best test scores based on the minimum MAE are considered. Hyperparameters are optimized for each model using a grid search with values for the learning rate in $\{0.001, 0.0003, 0.0001\}$ and the number of layers in $\{1, 2, 4, 8\}$. Each combination is repeated for three random seeds. The minimum average validation score is used to select the hyperparameters for reporting on the test set. The minimum average train scores are directly selected for reporting. ZINC is available under the license DbCL.

### A.3    HYPERPARAMETERS

For ZINC, a grid search on the learning rate in $\{0.001, 0.0003, 0.0001\}$ and the number of layers in $\{1, 2, 4, 8\}$ is performed for each method. For the function approximation, a grid search on the learning rate is performed with values $\{0.03, 0.01, 0.003, 0.001\}$. Selected hyperparameters are presented in Table 3 and Table 4.

| Method | Loss |
|---|---|
| GCN | 0.01 |
| GAT | 0.03 |
| GATv2 | 0.01 |
| SAGE | 0.03 |
| GIN | 0.003 |
| MC-GIN+softmax | 0.01 |
| MC-GIN (k=1) | 0.01 |
| MC-GIN | 0.01 |
| MCGC | 0.001 |

Table 3: Best hyperparameters (learning rate (LR)) for Table 1.

| Method | Train | | Test | |
|---|---|---|---|---|
| | LR | Layers | LR | Layers |
| GCN | 0.0001 | 8 | 0.0001 | 8 |
| GATv2 | 0.0003 | 8 | 0.0003 | 8 |
| SAGE | 0.0003 | 8 | 0.0003 | 8 |
| GIN | 0.0003 | 8 | 0.0003 | 8 |
| MC-GIN | 0.0001 | 8 | 0.0001 | 8 |

Table 4: Best hyperparameters for the results in Table 2. Learning rate (LR) and number of layers (Layers).