# OpenReview forum: "Multi-Channel Graph Convolutions"
_ICLR.cc/2025/Conference — ICLR 2025 Conference Withdrawn Submission_

### Official Review · Reviewer_kVTk · 2024-10-29

**Soundness:** 2
**Presentation:** 1
**Contribution:** 1
**Rating:** 3
**Confidence:** 4

**Summary:**

The paper proposes a new GCN where each channel's convolution operation is performed independently, similar to Convolutional neural networks. The authors show all the steps required to derive the formulation in the spectral space and its approximation for deriving the message-passing formulation for a GCN and a modified version of GIN. Finally, the authors test their model on two datasets: ZINC and synthetic.

**Strengths:**

The idea is exciting and easy to understand. The authors did a great job in terms of clarity.

**Weaknesses:**

There is no related work section, and finding papers with similar titles and contents that should have been considered is straightforward. For example:

hang, X.; Wang, J.; Wang, R.; Wang, T.; Wang, Y.; Li, W. Multi-Channel Graph Convolutional Networks for Graphs with Inconsistent Structures and Features. Electronics 2024, 13, 607. https://doi.org/10.3390/electronics13030607

Zhai, Rui, Libo Zhang, Yingqi Wang, Yalin Song, and Junyang Yu. "A multi-channel attention graph convolutional neural network for node classification." The Journal of Supercomputing 79, no. 4 (2023): 3561-3579.

Meng, L., Ye, Z., Yang, Y. et al. DeepMCGCN: Multi-channel Deep Graph Neural Networks. Int J Comput Intell Syst 17, 41 (2024). https://doi.org/10.1007/s44196-024-00432-9

 How the proposed approach differs from or improves upon the multi-channel approaches in these works?

Moreover, the final formulation for an MC-CGN is similar to that performed by Roth et al. 2024: "In this work, we propose to split graphs into multi-relational graphs and operate MPNNs on these." However, the differences are not discussed, nor is this method included in the experimental evaluation. The authors should explicitly compare their approach to Roth et al. 2024, highlighting key similarities and differences and including it in the experimental evaluation.

The experimental evaluation is minimal, with just a synthetic dataset and Zinc, which cannot support the author's claims. The authors should consider the setup proposed in Roth et al. 2024.

**Questions:**

I would like for the authors to put their contribution in perspective with the current state of the art.

---

### Official Review · Reviewer_pzC1 · 2024-11-03

**Soundness:** 3
**Presentation:** 2
**Contribution:** 3
**Rating:** 6
**Confidence:** 4

**Summary:**

This paper proposes an extension of the popular GCN architecture to multiple channels. It relies on Fourier analysis and follows a derivation to that of GCN but for the multi-channel case. Then it offers a GNN, MCGC, based on this theoretical analysis and proves the 1-WL expressivity of the GNN. The utility of MCGC is demonstrated by an approximation experiment and strong results on the ZINC dataset.

**Strengths:**

1. Background on previous work is well-written and clear. It seems the authors are well-versed.

2. Injectivity results and derivation of results are correct as far as I can tell and are presented rigorously.

3. Important extension of a very popular graph learning paradigm with good empirical results.

**Weaknesses:**

1. The connection between the derivation of MCGC and the actual architecture is lacking. I understand it but it is not clear, please respond with a better explanation of the connection.

2. The submission has many typos (lines 336, 347, 151, etc.). This needs to be revised as I don't feel this is presentation-ready yet.

3. Proposition 3.3 relies on relatively old tools and there are much newer theoretical insights that not only work on countable spaces, but on continuous feature vectors, where this work is applied, see [Amir et al.]

4. The way the architecture is implemented in practice is not well explained and I did not understand the difference between the MCGC and MC-GIN variants.

5. The experiments are not sufficient: The function approximation is nice as a bonus and the real-world experiment is onyl done on one dataset. I would expect to see results on QM9 such as the classical GIN was measured and also compaison against GAT and GCN on experiments in the original papers.

 [Amir et al.] Amir, Tal, et al. "Neural injective functions for multisets, measures and graphs via a finite witness theorem." Advances in Neural Information Processing Systems 36 (2024).‏

**Questions:**

1. Don't the A's in the definition of MCGC have to be symmetric? I don't see it written.

2.  "a countable multiset" line 338 is not a valid definition. The multiset is finite (n elements) and each element comes from a space of countable size.

3. The proof of Proposition 3.3 is very uninformative and missing key details. The definition of the function f is not understandable. Please rewrite this proof much more rigorously.

---

### Official Review · Reviewer_pxae · 2024-11-04

**Soundness:** 3
**Presentation:** 3
**Contribution:** 1
**Rating:** 3
**Confidence:** 5

**Summary:**

The paper introduces an architecture for GNNs to be able to process more than one graph at a time. The proposed GNNs are based on multi-channel convolutions which are introduced in section 3. The paper is well-written and the ideas are clearly presented. However, this work carries little novelty, and the motivation for the work is poorly motivated. The numerical examples fail to show a clear benefit of the method here presented.

**Strengths:**

1. The paper is well-written.
2. The main ideas ideas of the paper are clearly presented.

**Weaknesses:**

1. The paper has very little novelty, multi-channel GNNs have been introduced long ago and in several contexts:

Multi-Channel Graph Neural Networks
Kaixiong Zhou et al.

Multi-Channel Graph Neural Network for Entity Alignment
Yixin Cao et al.

DeepMCGCN: Multi-channel Deep Graph Neural Networks
Lei Meng et al.

The authors should explain what is new in this work.

2. Multi-channel graph convolutions are an example of polynomials on multi-graphs with degree 1:

Convolutional Filters and Neural Networks with Non-Commutative Algebras
A Parada-Mayorga, L Butler, A Ribeiro
IEEE Transactions on Signal Processing

Convolutional learning on multigraphs
L Butler, A Parada-Mayorga, A Ribeiro
IEEE Transactions on Signal Processing 71, 933-946

I fail to understand the benefit of the proposed methodology versus more general architectures. The authors should explain why their method (which is a particular case of multigraph convolutions) is both novel and beneficial.

**Questions:**

I will encourage the authors to make a convincing argument for the novelty of their method. Also, I would encourage the authors to consider a numerical example in which the benefits of the proposed methodology are seen.

---

> ### Author Response · Authors · 2024-11-15
>
> We thank the reviewer for their feedback and will clarify all points for resubmissions. Below, we provide comments on the misunderstandings.
>
> **1. The paper has very little novelty, multi-channel GNNs have been introduced long ago and in several contexts:**
> * We refer to the general comment above. These works refer to the term multi-channels as multiple different subgraphs, which is unrelated to our work.
>
>
> **2. Multi-channel graph convolutions are an example of polynomials on multi-graphs with degree 1:**
> * The opposite is true. Polynomials of any degree are specific filters that can be used within the general convolution as filter $\mathbf{W}$. Given $\mathbf{X}$ (assuming all signals are present) and $\mathbf{Y}$, the MCGC in their general form can express any such linear mapping $(\mathbf{W}* \mathbf{X}) = \mathbf{Y}$. Any MIMO polynomial (any degree) over a given simple graph can be expressed as a MCGC over that simple graph with a corresponding filter $\mathbf{W}\in\mathbb{R}^{n\times c\times d}$. These polynomials then approximate the general form but constrain the functions that can represented.
> * Polynomials over multi-graphs are specific filters for convolutions over multi-graphs. However, the general form of convolutions over multi-graphs has not yet been proposed.
> * Polynomials on multi-graphs with degree 1 are examples of our defined localized MCGC, which is indeed quite interesting.

---

### Official Review · Reviewer_dXbM · 2024-11-04

**Soundness:** 2
**Presentation:** 2
**Contribution:** 2
**Rating:** 3
**Confidence:** 3

**Summary:**

This paper introduces multi-channel data mixing for convolution-based graph neural networks. To achieve this, the authors propose Multi-Channel Graph Convolutions (MCGCs), which use the convolution theorem and graph Fourier transform to manage multi-feature node data—unlike conventional MPNNs, limited to single-channel signals. This multi-channel design allows MCGCs to create richer embeddings by amplifying distinct signals across channels. Building on this, the authors develop a scalable approximation, the Multi-Channel Graph Isomorphism Network (MC-GIN), which retains GIN-level expressivity (1-WL) while improving scalability. Experimental results indicate MCGCs can capture complex functions more effectively, although they may risk overfitting as task complexity grows.

**Strengths:**

- Innovative (even if the extension from scalar to vector data in graph convolution seems natural) and promising approach, with potential for further exploration. The concept of mixing node channels to form layer-specific learned representations is interesting, and the initial results look promising.
- The paper presents a clear and coherent storyline, making it accessible.
- Theorem 3.2 is a relevant extension of GIN's expressivity properties

**Weaknesses:**

- The authors frame the paper as offering a promising direction for improving convolution-based GNN architectures, yet more research is required on alternative approximations (e.g., MC-GIN) with better justification.
- Despite promising, the work is still preliminary and would benefit from further empirical validation; it’s only tested on a simple and carefully designed synthetic case and one real-world dataset. More comprehensive experiments are needed.
- The proposed method introduces new hyperparameters (e.g., the number of channels in intermediate layers), yet no strategy or guidance is offered for tuning these to avoid overfitting, as noted in the conclusion.
- Due to the additional parameters, this model incurs extra computational overhead. This point needs more clarification, discussion, and eventually empirical support by reporting runtimes.
- Since runtime is crucial in model selection, the overhead imposed by MCGC and derived architectures (e.g., MC-GIN) should be discussed and illustrated with more datasets.
- Figure 2 is somewhat cluttered and challenging to interpret; reducing input/output channels or nodes/eigenvalues could improve clarity.
- Figure 3 needs better formatting. Tables 1 and 2 would benefit from similar refinements. In these tables, runtime variability is not reported.

**Questions:**

- Theorem 3.1 appears to be presented as a theorem, but it reads more like a mathematical definition of multi-channel convolution. Could the authors clarify this distinction? Is this an actual contribution of this paper?
- Beginning of section 3.1: can the authors elaborate on what are the different $\hat{A}^{(i)}$?
- Can the authors elaborate on the interplay between Theorem 3.2 and Figure 3? While the model shares GIN's expressivity, it is unclear why it outperforms for so much the single-channel GIN in simple function approximation. This difference in function approximation within the same class of expressivity is worth some discussion after Theorem 3.2.
- Is there a straightforward extension for multi-channel edges?

---

### Official Review · Reviewer_F4hV · 2024-11-05

**Soundness:** 2
**Presentation:** 2
**Contribution:** 2
**Rating:** 3
**Confidence:** 4

**Summary:**

Previous graph convolutions are generally defined on single-channel input signals, which are then applied to all channels in spectral GNNs. This paper proposes a definition of graph convolutions on multi-channel signals (**MCGCs**).

MCGCs are computationally difficult to implement, so the authors further propose an approximation of MCGCs called **MC-GINs**, allowing for localized computation. Before deriving MC-GINs, the authors point out that GAT+softmax cannot be directly applied to various graph structures, as it would make the graph row-stochastic. Based on such careful considerations with MC-GINs' architecture, the authors provide an expressiveness analysis.

In the experiments, the authors validate the function-fitting ability of MC-GINs. They also conduct graph classification experiments, comparing MC-GINs with several classic baseline models.

**Strengths:**

1. The authors' analysis and extension of the existing graph convolution definition is commendable.

2. Thm. 3 (although I still think it should be labeled as a Definition) is explained in detail, even though there are some typos.

3. The first set of experiments shows that using GAT+softmax is not suitable, which supports the previous analysis.

**Weaknesses:**

1. I believe the biggest issue is that the transition from MCGCs to MC-GINs is quite unconvincing. The authors mention the perspective of multiple edge relations, but **it's still too unclear what the relationship between Eq. 2 and Eq. 3 is**. The authors claim that MCGCs are computationally impractical, but the graphs used in the experiments are quite small. It should have been possible to conduct experiments on MCGCs without deriving MC-GINs.

2. Regarding **"Theorem 3.1"**, the authors state: "We emphasize that this form is a mathematical fact and not a definition made by us." However, I still consider this to be a definition by the authors: to handle multi-channel signals, couldn't we simply learn different filter functions for each channel separately? I think this is also a type of definition—similar to what JacoviConv and OptBasisGNN do.

3. **TYPO**: In the third block equation below Eq. 2, the tensor shapes on both sides of the element-wise multiplication do not match.

4. The **baselines** used in the second set of experiments are quite simple. Given the motivation of the paper, it seems more appropriate to compare against Spectral GNNs.

5. The authors emphasize that MC-GINs should be **localized**, but Proposition 3.3's **implementation of $f$ is not localized**. Because of this, the graphs used in the experiments are all small.

**Questions:**

Please refer to the Weaknesses section.

---

> ### Author Response · Authors · 2024-11-15
>
> We thank the reviewer for their feedback and will clarify all points for resubmissions. Below, we provide comments on the misunderstandings.
>
> **Regarding "Theorem 3.1", the authors state: "We emphasize that this form is a mathematical fact and not a definition made by us." However, I still consider this to be a definition by the authors: to handle multi-channel signals, couldn't we simply learn different filter functions for each channel separately? I think this is also a type of definition—similar to what JacoviConv and OptBasisGNN do.**
>
> * While the terms convolution and filter are often used interchangeably in graph machine learning, they are fundamentally different. The convolution is the operation $\mathbf{W} * \mathbf{X}$ with $\mathbf{W}$ being the filter. It has the clear definition that it is the inverse Fourier transform of the pointwise multiplication in Fourier space. For a given multi-channel signal (a matrix) it has precisely the form that we have shown in Theorem 3.1. The only definition used here is the definition of the convolution in general.
> * JacoviConv, OptBasisGNN (and any other filter) can be seen as defining specific filters $\mathbf{W}$ within the MCGC. In the general form (Theorem 3.1), the MCGC can map any given $\mathbf{X}$ to any given $\mathbf{Y} = \mathbf{W}*\mathbf{X}$ with a suitable filter $\mathbf{W}$. Any specific filter (JacoviConv, OptBasisGNN,...) is strictly less expressive. By knowing the general form of the MCGC, we can find better approximations, i.e., better filters.
>
> **TYPO: In the third block equation below Eq. 2, the tensor shapes on both sides of the element-wise multiplication do not match.**
>
> * We compute $\mathbf{A} \odot \mathbf{B}$ where $\mathbf{A}\in\mathbb{R}^{n\times c\times d}$ and $\mathbf{B}\in\mathbb{R}^{n\times d}$ where $d$ is the number of input and $c$ the number of output channels. With $\odot$, the matrix-vector product $[\mathbf{A} \odot \mathbf{B}]_i = \mathbf{A}_i\mathbf{B}_i$ is performed where $\mathbf{A}_i\in\mathbb{R}^{c\times d}$ and $\mathbf{B}_i\in\mathbb{R}^{d}$.
>
> **The authors emphasize that MC-GINs should be localized, but Proposition 3.3's implementation of is not localized. Because of this, the graphs used in the experiments are all small.**
> * Proposition 3.3 states that MC-GIN is injective on multisets $\mathbb{X}_i\subset \mathbb{X}$. In message-passing, these multisets correspond to the multisets of neighboring node states $\{\{h_j | v_j \in N_i \}\}$. Thus, MC-GIN aggregates messages from the local neighborhood of each node as in other MPNNs.

---

### Official Review · Reviewer_3Lkf · 2024-11-05

**Soundness:** 1
**Presentation:** 2
**Contribution:** 1
**Rating:** 1
**Confidence:** 5

**Summary:**

The paper purports to define graph neural networks with multiple channels. Graph neural networks with multiple channels exist. They are actually standard in applications.

**Strengths:**

None

**Weaknesses:**

The paper proposes a learning architecture that already exits in the literature.

**Questions:**

None

---

### Official Review · Reviewer_stEr · 2024-11-09

**Soundness:** 2
**Presentation:** 1
**Contribution:** 2
**Rating:** 3
**Confidence:** 3

**Summary:**

The authors propose a multi-channel graph convolution (MCGC) based on the convolution theorem and the graph Fourier transform, specifically designed for multi-channel signals. In this context, the authors also introduce the multi-channel graph isomorphism network (MC-GIN), which provides a linear map with the same expressive power as the Graph Isomorphism Network (GIN).

**Strengths:**

The mathematical definition of convolution for multi-channel signals is very interesting and offers great potential; the authors, for example, used it as a basis for MC-GIN.

**Weaknesses:**

Here is a list of points in the paper that do not convince me  (not in order of importance):
- I see no actual discussion regarding complexity (I may have missed it); only MAE-related times are reported, which are not very indicative.
- I find the introduction to the multi-channel concept confusing right from the initial definitions. For greater clarity, I would suggest also explaining the difference between multi-in multi-out and single-in multi-out from the beginning and perhaps providing clearer examples
- Unfortunately, although the paper is ok in many sections, the experimental part is truly lacking. Not only is the presentation of the experiments unclear, but the experiments themselves are also insufficient.
- While the idea of the Fourier transform for graphs is valid, it is unclear how innovative it truly is, which also affects the scientific contribution.
- The strengths of the work have not been clearly highlighted.

**Questions:**

- Figures 1 and 3 would require more detailed explanations in the caption, as they are currently difficult to understand.
- I think the definition “For d = 1, we refer to this as a single-channel signal, while for d > 1, we call it a multi-channel signal” may create confusion in the reader, as it seems to refer to the dimensionality of the feature. This is an essential point and would benefit from greater clarity.
- I would clearly explain the difference with multiple attention heads.

---

### Author Response · Authors · 2024-11-15

We thank all reviewers for their valuable feedback, which will help us improve our work for resubmission. As reviewers misunderstood various parts of our work, we take the opportunity to clarify these points for future readers.

Various reviewers pointed out existing works on multi-channel graph neural networks [1,2,3,4,5]. While the title of these works is similar to ours, this is mainly due to an overloading of the term multi-channel. All of these works use the term multi-channel to refer to multiple subgraphs, on each of which message-passing is performed. E.g., in Multi-Channel Graph Neural Networks [1], multiple instances of DiffPool construct different pooled graphs, each of which is referred to as a channel. They then propose to learn interactions (an adjacency matrix) between nodes in these different subgraphs. Contrarily, our work considers a single graph, where multi-channels refer to feature channels as defined in graph signal processing, i.e., the graph signal is a function that maps each node of the graph to a d-dimensional vector.

We clarify further points as individual comments.

---

[1] Zhou et al., Multi-Channel Graph Neural Networks, IJCAI (2020).

[2] Cao et al., Multi-Channel Graph Neural Network for Entity Alignment, ACL (2019).

[3] Meng et al., DeepMCGCN: Multi‑channel Deep Graph Neural Networks, IJCIS (2024).

[4] Chang et al., Multi-Channel Graph Convolutional Networks for Graphs with Inconsistent Structures and Features, Electronics (2024).

[5] Zhai et al., A multi‑channel attention graph convolutional neural network for node classification, Supercomputing (2022).

---

> ### Comment · Reviewer_3Lkf · 2024-11-21
>
> Your statement that your "work considers a single graph, where multi-channels refer to feature channels as defined in graph signal processing, i.e., the graph signal is a function that maps each node of the graph to a d-dimensional vector" is well understood.
>
> Unfortunately, it is standard to use GNNs with multiple channels defined in this way.

---

### Note · Authors · 2024-11-25

I have read and agree with the venue's withdrawal policy on behalf of myself and my co-authors.